# Physical Exercise Decreases the Mobile Phone Dependence of University Students in China: The Mediating Role of Self-Control

**DOI:** 10.3390/ijerph16214098

**Published:** 2019-10-24

**Authors:** Guan Yang, Guang-xin Tan, Yue-xiang Li, Hai-ying Liu, Song-tao Wang

**Affiliations:** 1School of Physical Education and Sports Science, South China Normal University, Guangzhou 510006, China; tangx33@bu.com; 2Department of Anthropology, Boston University, MA 02215, USA; 3Department of Physical Education, Guangzhou Vocational and Technical University of Science and Technology, Guangzhou 510550, China; Liyuexiang@gvtusc.edu.cn; 4School of Physical Education, Guangzhou University, Guangzhou 510006, China; liuhy2016@gzhu.edu.cn

**Keywords:** physical exercise, mobile phone dependence, self-control, mediating effect, university students, China

## Abstract

This study aimed to explore the relationship between physical exercise (PE) and mobile phone dependence (MPD) in Chinese university students and verify the potential role of self-control (SC) in mediating the decrease of MPD by PE. Through the quota sampling, 608 students that complied with the requirements were surveyed from 10 universities in China. PE, MPD, and SC were assessed using standard scales. For data analysis, t-tests, correlation analysis, hierarchical regression, and regression analysis were carried out in turn. The results showed significant gender differences in PE, MPD, and SC. For male students, the level of PE and score of SC were higher than those in females, yet the MPD score was lower. PE and SC were negatively related to MPD (*r* = −0.124, *p* < 0.01; *r* = −0.563, *p* < 0.001), so both could remarkably predict MPD (*β* = −1.00, *p* < 0.05; *β* = −0.552, *p* < 0.001). Gender was also a significant predictor for MPD (*β* = 0.089, *p* < 0.05). PE could, to some extent, decrease MPD, in which SC played a significant mediating role—its mediating effect accounted for nearly 71% of the total effect. The present study shows that PE is negatively correlated with MPD, and SC mediates the decrease of MPD by PE for university students in China. This indicates that the improvement of SC by PE could be a highly trustworthy and practicable way to effectively address the issue of MPD in university students or other young people across the world.

## 1. Introduction

Today, the rapid development of the Internet and other types of information and communication technology has brought exponential growth to mobile phone users [1,2]. In many applications of mobile phones, especially smart phones, functions related to the Internet have become the principal use of mobile phones, replacing traditional calling and information transmission methods. According to the latest data from the 43rd statistics report in 2019 put forth by the China Internet Network Information Center (CNNIC), the number of the mobile phone netizens has already reached 0.817 billion, and the percentage of those with access to the Internet via mobile phones has even reached 98.6% of total users [3]. Among mobile phone netizens, it is apparent that young adults (aged 18–25 years) have become the largest group of users, making up an incredible ratio of more than one third of users. The majority among this group might be surmised to be university students. As a very practical and multifunctional digital device, mobile phones are widely employed by university students. 

They apply mobile phones to various kinds of aspects in daily life, including shopping online, playing games, watching videos, browsing journals, communicating with others and instant buying or paying [4]. However, it should be considered that excessive mobile phone usage might do more harm than good and is even likely to cause some serious passive consequences, such as mobile phone dependence (MPD) [5]. MPD has also been called problematic mobile phone use or mobile phone addiction [6,7]. As for university students, mobile phone overuse not only leads to a poor academic performance [8] but can also contribute to some mental disorders such as depression, social anxiety, stress, and insomnia, as well as negative emotions in general [9,10,11].

Apart from the detrimental effects on learning and mental health, excessive mobile phone use can to a large extent disrupt university students’ physical activity [12], as supported by several previous findings [13,14,15]. Kim et al. found that smart phone addiction in university students may negatively influence physical health by reducing the amount of time spent in physical activity such as walking [13]. Penglee et al. revealed that high smart phone use among college students may be a barrier to physical activity and proposed a strategy for physical activity promotion in higher education settings [14]. Similarly, Barkley and Leep pointed out that mobile phone used by college students can be viewed as a sedentary leisure behavior resulting in poor physical activity [15]. Therefore, from these above studies, it is not difficult to surmise that PE and MPD may be inversely related to each other. 

Furthermore, several personality traits have been found to be associated with the extent of mobile phone usage, such as self-esteem, self-regulation, and self-control (SC) [16,17]. Compared to self-esteem and self-regulation, SC has been verified to be a more important psychological predictor for problematic mobile phone use, showing a negative association between the two [18]. High self-control has been proved to be a highly indispensable protection factor against Internet addiction in adolescents. Nevertheless, facing stimuli and temptation from the visual world, youth with low self-control are extremely prone to Internet addiction [19]. Considering the likely correlation between SC and MPD, the improvement of SC might imply a decrease of MPD. At this point, however, the key problem is determining how can we effectively improve SC. 

In accordance with the strength model of SC, the strength depletion of SC might resemble the use of muscle, which can become strong by regular exercise or recover again with proper rest after being used for a period of time [20,21]. In other words, just as muscle strength can be naturally enhanced by exercise or training, perhaps SC can be improved in the same way. If SC could be effectively improved or enhanced via regular physical activity or exercise, similar to the growth of muscle, PE might be more likely to have a positive relation to SC. It is fortunate that, to date, some previous studies have proven the positive relationship between PE and SC, including different intensities [22,23], different periods [24,25], and different types physical activity and exercise programs [25,26]. Bibley et al. found that moderate intensity aerobic exercise could improve the inhibitory function of university students, an indispensable component of SC [23], but Kamijo et al. pointed out that high intensity physical exercise may not strengthen individuals’ SC [22]. Both Tomporowski et al. and Davis et al. indicated that both acute and chronic physical exercise can be beneficial to the enhancement of SC [24,25]. In addition, Davis et al. suggested that group activities such as football or basketball can also effectively improve SC [25].

In summary, although some previous studies have revealed a negative relationship between MPD and PE or SC, more potent and reliable evidence in needed to confirm these findings. Thus, we investigated whether frequently participating in physical activity or exercise could decrease MPD of university students, in which SC plays an important mediating role. It is fortunate that the relation between exercise rehabilitation and smart phone addiction has gained attention around the world [27]. Therefore, the current study aimed to examine the relationship between PE and MPD in Chinese university students and to verify the mediating role of SC in the decrease of MPD by PE, as well as the association of those factors with demographic variables, such as gender, major, and hometown. At the same time, we also hypothesized that a negative relationship exists between PE and MPD and that SC plays a crucial mediating role in the decrease of MPD by PE.

## 2. Materials and Methods 

### 2.1. Procedure and Participants

This study was performed in accordance with the Declaration of Helsinki and approved by the institutional review board of the South China Normal University in response to the involvement of minimal risk and anonymous survey procedures. It was also supported by the Innovation Project of Graduate School of South China Normal University (2018LKXM011). All subjects gave written informed consent before participating in this survey.

A cross-sectional survey was conducted using several self-reported standard scales. Through the quota sampling between 1 December 2018 and 26 January 2019, about 60 to 70 students were selected from each of the 10 studied universities in Guangzhou, China. A total of 650 university students were surveyed. The survey was completed anonymously and confidentially, and all questionnaires were collected based on the voluntary principle. Under further examination, 42 samples were invalid due to missing data or incorrect/incomplete answers.

### 2.2. Measures and Instruments

PE was measured by the Physical Activity Rating Scale-3 (PARS-3) [28]. The PARS-3 is a three-item self-reported scale, containing exercise intensity, exercise time, and exercise frequency. Each item is rated from 1 to 5, and the total score of physical activity is computed by the following equation: intensity × (time−1) × frequency, with a range of 0 to 100. A total score that is equal to or less than 19 is defined as light, one that is 20 to 42 is defined as moderate, and one that is equal to or more than 43 is defined as vigorous physical activity. According to previous experience [29], the current study divided the light physical activity into two components: no and light physical activity. No physical activity is equal to or less than 4, and the light physical activity ranges from 5 to 19. Thus, PE in this study was divided into four levels, from 1 (no physical activity) to 4 (vigorous physical activity). The PARS-3 has excellent test–retest reliability (*r* = 0.82). The internal consistency of PARS-3 in this study was basically satisfactory, and the Cronbach’s α was 0.639. 

The MPD was assessed by the mobile phone addiction tendency scale (MPATS) [30], which is based on Young’s internet addiction scale [31]. The MPATS is a five-point Likert scale with 16 items and four dimensions: withdrawal symptom, salient behavior, social comfort, and mood change. Each item is scored from 1 (completely disagree) to 5 (completely agree), and the total score has a range of 16 to 80. A higher score indicates a deeper degree of MPD. Both exploratory and confirmatory factor analyses have supported the construct validity of the four dimensions. The internal consistency coefficient of MPATS is 0.83, and the test–retest reliability is 0.91 [30]. Additionally, a previous study demonstrated that this scale was conducted well in a sample of college students [32]. The Cronbach’s α of the present study was 0.895.

The SC was evaluated by the self-control scale (SCS) [33], which was modified based on Tangney’s Self-Control Scale [34]. The SCS is a five-point Likert scale and comprises 19 items. It has five dimensions: controlling impulses, keeping healthy habits, resisting temptation, concentrating on work and controlling entertainment. Each item is valued from 1 (completely disagree) to 5 (completely agree). The total score can be from 19 to 95, and a higher score shows a higher level of individual self-control. The SCS has a fair internal consistency coefficient (α = 0.862) and test–retest reliability (*r* = 0.850). Moreover, a previous study proved that this scale was fairly in a sample of college students [35]. The Cronbach’s α in this study was 0.891.

Additionally, several basic demographic variables, including gender, major, and hometown, were recorded on a standard survey form.

### 2.3. Statistical Analysis

All statistical analyses were performed with IBM SPSS statistical software version 21.0 for Windows (SPSS, Chicago, IL, USA). Continuous variables with normal distribution were presented as the mean ± standard deviation (SD), and categorical variables were displayed as number and percentages (%). Independent sample t-tests were used to determine any possible differences of PE, MPD, and SC, respectively, in related groups based on demographic variables. As the level of PE was measured as an ordinal variable, Spearman’s correlation analysis was conducted to examine the relationship between PE and MPD or SC. The relationship between MPD and SC was calculated by Pearson’s correlation analysis. Hierarchical regression analyses were used to determine the variance of MPD explained by gender, major, hometown, PE, and SC. 

Finally, in order to verify the mediating effect of SC between PE and MPD, according to previous experience [36,37], the following three points were determined through a series of regression analyses: (1) the independent variable (PE) was found to be significantly correlated to a dependent variable (MPD), (2) the independent variable was found to have a significant relation to the mediator (SC), and (3) the mediator was found to be highly associated with the dependent variable. If the above points were all met, the mediating effect surely existed. However, depending on whether the SC entirely mediates the relationship between PE and MPD, the direct effect of PE on MPD may not be significant. Otherwise, a partial mediating effect would be presented. The significance level of the present study was set at *p* < 0.05.

## 3. Results

### 3.1. Sample Characteristics Analysis

As shown in Table 1, the sample in this study was 608 undergraduates, and its overall response rate was 93.53%. The sample consisted of 158 males (25.99%) and 450 females (74.01%) students. In the sample, 317 students (63.98%) were liberal arts majors, and 291 (36.02%) were science majors. In terms of hometown, 455 students (25.16%) were from urban areas, and 153 (74.84%) came from rural areas. The means of PE, MPD and SC levels was 2.12, 42.81 and 59.69, respectively.

The t-tests showed that male students had a significantly higher PE level than female students (*t*(606) = 10.833, *p* = 0.000, *d* = 1.002), and science students also had a significantly higher PE level than liberal arts students (*t*(606) = −3.397, *p* = 0.001, *d* = 0.276). Female students’ MPD scores were significantly higher than those of male students (*t*(606) = −2.948, *p* = 0.003, *d* = 0.273), and liberal arts students’ MPD scores were also significantly higher than those of science students (*t*(606) = 2.742, *p* = 0.006, *d* = 0.223). No significant difference was found in levels of PE and scores of MPD between rural and urban areas. Likewise, male students’ SC scores were higher than those of female students (*t*(606) = 2.568, *p* = 0.01, *d* = 0.237). However, no significant differences were revealed in the SC scores between major and hometown groups.

### 3.2. Spearman’s or Pearson’s Correlation Analysis

As shown in Table 2, the correlation coefficients of PE, MPD, and SC were all statistically significant. The PE was negatively correlated with MPD (*r* = −0.124, *p* < 0.01) but positively associated with SC (*r* = 0.164, *p* < 0.001), and the correlations were all low. The MPD was inversely related to SC (*r* = −0.563, *p* < 0.001), and the correlation was high.

### 3.3. Hierarchical Regression Analysis 

Table 3 exhibits the hierarchical regression analyses that were performed to identify the relative explanatory variance of the three sets with demographic indicators as well as the effects of PE and SC on MPD. Gender, major, and hometown were first introduced in Block 1, then PE followed in Block 2, and SC entered last and is depicted as Block 3. For each model, the change of *R*^2^ and *F* is given.

It can be seen that the percentage of *R*^2^ increased from 2.2% to 32.5%, and each block showed significant explanatory variance to MPD (see Table 3). In Model 1, gender positively and significantly predicted MPD (*β* = 0.089, *p* < 0.05), accounting for 2.2% of the variance of MPD. The PE emerged as a remarkably negative indicator in Model 2 (*β* = −1.00, *p* < 0.05), explaining 0.8% of the variance of MPD. The final step added SC of Block 3 to Model 3, which could significantly and negatively predict MPD (*β* = −0.552, *p* < 0.001), accounting for 29.5% of the variance of MPD. In all three models, major and hometown barely reached the significant level of 5% to forecast MPD, and these results suggests that SC could be a negative predictor for MPD, which was also in line with the Pearson’s correlation analysis result.

### 3.4. Mediating Role Analysis

In order to test the mediating role of SC between PE and MPD, three steps had to be performed in sequence. The detailed steps and results are represented in Table 4.

In Step 1, the regression analysis of MPD on PE explained 1.7% of variance in MPD and the model had a good fit (*F* (1,606) = 10.643, *p* < 0.01), which means that PE was a significant predictor of MPD (*β* = −0.131, *p* < 0.01). In Step 2, the regression analysis of SC on PE revealed 2.8% of variance in SC and the model had a fair fit (*F* (1,606) = 17.598, *p* < 0.001). This means that PE could significantly and positively predict SC (*β* = 0.168, *p* < 0.001). In Step 3, the regression analysis of MPD on SC and PE together interpreted 31.8% of the variance in MPD. Likewise, the model had an excellent fit (*F* (2,605) = 141.215, *p* < 0.001). All of the above analyses showed that SC was in fact a mediator between PE and MPD and completely controlled the decrease of MPD by PE. The mediating effect of SC accounted for nearly 71% of the total effect.

## 4. Discussion

The objective of this study was to examine the relationship between PE and MPD among university students in China and to certify the mediating role of SC in the decrease of MPD by PE, as well as examine the association of those factors with demographic variables. The present study revealed that male students had a significantly higher PE level than female students, and science students’ level of PE was also significantly higher than that liberal arts students. Female students’ MPD score was significantly higher than that of males, and liberal arts students’ MPD score was also significantly higher than that of science students. MPD was negatively associated with PE and SC, yet PE was positively associated with SC. Moreover, gender was an effective predictor for MPD, whereas PE and SC were found to negatively predict MPD. These findings were in line with the results of the correlation analyses among PE, MPD and SC. Most importantly, PE could negatively affect MPD directly as well as indirectly through SC, which means that SC was a mediator between PE and MPD. However, when controlling for SC, the direct effect of PE on MPD was not significant, which implies that SC played a complete mediating role in the course that PE decreased MPD. The findings obtained from this study confirmed our hypothesis and provided more genuine and reliable evidence to identify the relationship between PE, MPD, and SC. 

According to previous studies [6,35,38], gender is a very important demographic variable to distinguish the differences in MPD among university students. Likewise, this study revealed that female students were more likely to be addicted to mobile phones, supporting the finding that gender is a significant variable that can be used to predict MPD. Different use patterns of mobile phones can explain why female students’ MPD was greater than that of their male counterparts. For instance, females are inclined to apply mobile phones to establish and maintain interpersonal relationships, but males prefer to talk in a face-to-face manner instead of using mobile phones [39]. Additionally, females shop online via mobile phones more frequently than males [35,39]. Compared to male students, the present study found that females lacked enough PE. Given the results of the current study, spending more time on mobile phones for female students inevitably affected physical activity directly; in other words, compared to their male counterparts, females participated in less physical activity. As previous research has suggested [14,40], the increase of screen time and sedentary behavior caused by mobile phone overuse among female students indirectly gives rise to a lack of adequate physical activity in daily life. Furthermore, the male students’ SC scores were considerably stronger than those of the female students in this study, which is somewhat different from the findings of previous literature [35]. As an example, Jiang and Zhao found that male students’ scores were higher than the female, but there was no statistically significant difference between genders [35]. Therefore, we speculate that the possible cause might be associated with the level of PE or the degree of MPD among male and female students in this study. 

Consistent with the previous findings [12,13,14], the current study revealed that PE was negatively correlated with MPD, also being an effective predictor for MPD. Kim et al. found that students with high-level MPD usually showed less physical activity on a day to day basis, based on the total number of walking steps [13]. Rebold et al. pointed out that students, in mounting numbers, are willing to employ sports apps on mobile phones, but this might reduce the intensity of physical activity [41]. At the same time, they also found that mobile phone use could disrupt physical activity, such as reducing runners’ heart rates [42]. Furthermore, spending more time on mobile phones leads to an increase in sedentary behavior, possibly indicating that less time is invested in PE [12,15]. Apart from PE, this study found that SC was also inversely related to MPD and negatively predicted MPD, which is in agreement with most previous findings [18,35]. Khang et al. reported that those with low self-control are easily susceptible to the pathological use of mobile phones, implying that a direct association may exist between SC and mobile phone overuse [17]. As previous research has suggested [34,43], SC is a highly essential ability to delay gratification. Namely, with the purpose of accomplishing long-term plans or goals, one must suppress individual impulsiveness and give up immediate and small rewards. When facing temptation from mobile phones, if there is a lack of ability to inhibit impulsiveness in time, people may lead to misconduct such as problematic or excessive mobile phone use. Mei et al. pointed out that a significant positive correlation existed between impulsiveness and mobile phone overuse [44].

In agreement with previous studies [45,46,47], the present study found that PE had a positive relation to SC, which means that PE could to some extent improve university students’ SC. In other words, the higher level of PE they had, the stronger SC they will possessed. Just like the statement discussed in the introduction, the reason why PE could strengthen individual SC might be interpreted by the theory of the strength model of SC, which points out that SC is similar to the use muscle. It relies on a limited energy resource and might be temporarily exhausted by continuous usage [20,21]. As an example, even though the strength of a muscle would be temporarily depleted in resistance exercise, it will quickly recover again and become stronger after a period of proper rest; during this period, the SC could be also improved [45]. Similarly, during aerobic exercise, especially the long-distance running or swimming programs, it may be difficult to persist until the end, and some may be more likely to choose to give up in the middle. However, as long as an individual can conquer the urge to give up and achieve the endpoint, the willpower or resilience equal to SC can also be strengthened [46,47]. Therefore, we believe that behaviors that require exercise of willpower or resilience are likely to improve individual SC [48,49], which can provide theoretical support demonstrating the key role of SC between PE and MPD. Apart from PE, some other methods could also be applied to improve SC, such as controlling emotions or diet [50], keeping learning habits [51], and avoiding impulsive buying behaviors [52]. 

The most important finding in the current study was that SC was a mediator between PE and MPD. SC entirely mediated the effect of PE on MPD, with the mediating effect of nearly 71%. As a matter of fact, a complex relationship probably exists in the manner that PE decreases MPD, because PE could both directly influence MPD and indirectly affect MPD through the mediator of SC. The mediating course can be divided into two steps: first, with improvement of the level of PE, the ability to SC enhances naturally; second, due to the increase of SC, the phenomena of excessive or problematic mobile phone use could decrease, by which the degree of MPD would be attenuated. The first step can be proved and interpreted by the strength model of SC [20,43,44], and the second step could be explained and testified by this study and the previous researches discussed above [17,35,43]. Thus, the improvement of SC is undoubtedly an effective and practicable means by which university students can cope with the problem of MPD.

In addition, on the basis of the dual-systems model of SC first put forward by Hofmann and Strack [53], the majority of behavioral problems among adolescents can be attributed to the imbalance between impulsive and self-control systems (i.e., under the dominance of the impulsive system, young people are more likely to generate dependent behaviors) [54,55]. Therefore, determining how to effectively inhibit the impulsive system and improve the self-control system may be a feasible path to reduce dependent behaviors in adolescents [56,57]. As previous findings have suggested [46,51], regular and chronic physical activity or exercise could reduce risk, impulsiveness, and other problematic behaviors and help individuals to develop and maintain decent behavior habits and healthy lifestyles. Consequently, relying on inhibiting the impulsive system and improving the self-control system, PE could decrease excessive or problematic mobile phone use so that MPD can be attenuated gradually among university students. 

Although SC plays a quite imperative mediating role between PE and MPD, PE itself can also directly reduce the MPD of university students to a certain degree. First, actively participating in physical activity and exercise programs might contribute to cutting down the screen time and sedentary behavior among university students so that they have less time to devoted to mobile phones and a lower chance of being addicted to it [12,14,58]. Second, it is generally acknowledged that mobile phones addiction is frequently accompanied by a series of adverse mental disorders, including depression, anxiety, stress, and negative emotions [9,10,11]. A growing number of research studies, however, have disclosed that PE may effectively promote mental health, especially through the attenuation of symptoms of anxiety and depression and the improvement of mood [59,60,61]. Last but not least, some studies have revealed that problematic mobile phone users usually possess poor interpersonal relationship and low life satisfaction [38,62,63]. This is the reason why lots of university students abandon themselves to video games and social networks. However, PE can greatly improve individuals’ self-esteem and self-satisfaction, acquiring more social support to promote interpersonal communication and reduce the sense of loneliness in daily life [64,65,66].

Restricted by some subjective and objective conditions, this study inevitably contains a few limitations. First, we could not obtain causal results due to the cross-sectional design, so a longitudinal study design should be employed as soon as possible in the future. Second, all data were collected by self-reported, so some devices of artificial intelligence including 3D-Sensor Pedometer and sports bracelet should be employed to measure physical activity in the future so that we can acquire more accurate and quantified data, such as body mass index (BMI), walking steps, heart rate, and blood pressure. Third, more participants should be selected through random sampling, by which more genuine, reliable, and valid data could be acquired. Only in this way can we summarize more generalized conclusions. Additionally, the Cronbach’s α of PARS-3 was 0.639 and just reached the baseline level, but the scale has only three items that must be mainly responsible for this circumstance. Last but not least, apart from self-control, some other vital psychological indicators may also have an effect on the relationship between PE and MPD. Thus, other vital variables related to the link between PE and MPD should be taken into account in further studies. 

## 5. Conclusions

In the present study, significant gender differences were found in PE, MPD, and SC, implying that gender is a significant predictor for MPD. Additionally, an inverse proportion was found between PE and MPD, and SC played an important mediating role in the effect of PE on MPD, with the mediating effect of nearly 71% accounting for the most of total effect. We hope these findings in the present study could be helpful to state, government, and relevant departments of education to effectively deal with the troublesome MPD problem among university students and other young people in the near future.

## Figures and Tables

**Table 1 ijerph-16-04098-t001:** Comparisons of physical activity (PE), mobile phone dependence (MPD), and self-control (SC) in groups based on demographic indicators in university students.

Variable	Total	PE		MPD		SC	
Range		1−4		16−80		19−95	
	*N* (%)	*M* (SD)	*t*	*M* (SD)	*t*	*M* (SD)	*t*
All	608	2.12 (0.95)	*p*	42.81 (10.63)	*p*	59.69 (10.87)	*p*
Male	158 (25.99)	2.77 (1.05)	10.833	40.68 (10.62)	−2.948	61.59 (11.63)	2.568
Female	450 (74.01)	1.89 (0.80)	0.000 ***	43.56 (10.54)	0.003 **	59.03 (10.52)	0.01 *
Liberal Arts	317 (63.98)	1.99 (0.86)	−3.397	43.94 (10.82)	2.742	59.09 (10.53)	−1.436
Science	219 (36.02)	2.25 (1.03)	0.001 **	41.59 (10.29)	0.006 **	60.35 (11.20)	0.152
Rural	153 (25.16)	2.10 (0.93)	0.773	43.20 (10.69)	−1.528	60.60 (11.25)	1.194
Urban	455 (74.84)	2.17 (1.01)	0.440	41.68 (10.38)	0.127	59.39 (10.73)	0.233

Note: * *p* < 0.05; ** *p* < 0.01; *** *p* < 0.001.

**Table 2 ijerph-16-04098-t002:** Spearman’s or Pearson’s correlation coefficients of PE, MPD, and SC.

Variable	PE	MPD	SC
PE	――		
MPD	−0.124 **	――	
SC	0.164 ***	−0.563 ***	――

Note: ** *p* < 0.01; *** *p* < 0.001.

**Table 3 ijerph-16-04098-t003:** Demographic indicators, PE, and SC as predictors of MPD.

	Variable	Beta (*β*)	*t*	*ΔR* ^2^	*ΔF*
Model 1				0.022	4.427 **
Block 1	Sex	0.089	2.080 *		
	Major	−0.075	−1.757		
	Source	0.042	1.021		
Model 2				0.008	5.244 *
Block 1	Sex	0.049	1.054		
	Major	−0.075	−1.753		
	Source	0.043	1.063		
Block 2	PE	−1.00	−2.286 *		
Model 3				0.295	263.233 ***
Block 1	Sex	0.031	0.815		
	Major	−0.063	−1.757		
	Source	0.023	0.667		
Block 2	PE	−0.017	−0.452		
Block 3	SC	−0.552	−16.224 ***		

Note: Beta (β) = standardized coefficients; ΔR^2^
*=* R Square Change; ΔF *=* F Change; * *p* < 0.05; ** *p* < 0.01; *** *p* < 0.001.

**Table 4 ijerph-16-04098-t004:** Regression analysis to test the mediating effect of SC between PE and MPD.

	*B* (SE)	Beta (*β*)	*t*	*R*	*R* ^2^	Adjusted *R*^2^	*F*
Step 1				0.131	0.017	0.016	10.643 **
PE → MPD	−1.464 (0.449)	−0.131	−3.262 **				
Step 2				0.168	0.028	0.027	17.598 ***
PE → SC	1.915 (0.456)	0.168	4.195 ***				
Step 3				0.564	0.318	0.316	141.215 ***
SC → MPD	−0.544 (0.033)	−0.557	−16.344 ***				
PE → MPD	−0.422 (0.380)	−0.038	−1.112				

Note: B = unstandardized coefficients; SE = standard error; Beta (β) = standardized coefficients; * *p* < 0.05; ** *p* < 0.01; *** *p* < 0.001.

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
