# Peer review of "Physical Exercise Decreases the Mobile Phone Dependence of University Students in China: The Mediating Role of Self-Control"

_ijerph, 2019, doi:10.3390/ijerph16214098_

Round 1

Reviewer 1 Report

Please refer on effect sizes during statistical analyses. I wonder at least medium effect sizes were needed for saying statistical significant differences.  

Reviewer 2 Report

The English language needs to be refined to the grammar.
The relationship between PE and SC can be reciprocal. The paper mainly addressed only the mediated effect of SC, whereas it needs to be addressed in the mediated effect aspect of PE on the effect of SC. Therefore, references to previous studies on this part are needed in the introductory section.
In the discussion section (ppp277 to 288), the author mentions PE and SC's mechanism for reducing MPD, which may be addressed with other explanations.
SC is a high-level concept of decision making, and this means that people with higher levels of SC can balance their physical activity and mobile phone use.
Therefore, statistical analysis and analysis results should be discussed not only for the mediated effectiveness aspects of SC, but also for the mediated effectiveness aspects of PE.

Reviewer 3 Report

Thank you for this submission. While I feel this research reports on an important topic, this research study doesn't fit the current Journals mission. In addition, the grammar in the manuscript needs to reread and some parts should be rewritten. In addition, the overall scientific soundness is weak.

Abstract:

Line 15: Sentence does not make sense.

Introduction:

Line 45: Long and awkward sentence please reword.

Line 80: Change “the” to a, and change found to investigated.

Line 81: Please rewrite this sentence.

Methods:

Line 102-103.Please remove this last sentence. This should be reported in the results section.

Line 121: What do you mean conducted? Has it been validated in college age students?

Line 123. Why are you reporting Cronbachs A in the methods?

Line 139: Change “run” to “used”

Results:

Line 155: Remove “was”

Line 158 what are the units for PE, MPD, and SC?

Line 179: What do you mean rarely. This is confusing please reword.

Line 193: In the results you do not explain you only report what happen. Please move to the discussion.

Line 206. In the results you do not explain you only report what happen. Please move to the discussion.

Discussion:

Please restate the purpose of the study for the readers.

Line 212: Please reword this sentence.

Line 224: Please change “sex” to gender.

Line 225: Please rewrite this sentence.

Line 226-227: A reference is needed to support this statement.

Line 233-234: Please reword this sentence.

Line 241: Please reword this sentence.

Line 257: Please reword this sentence with correct grammar.

Round 2

Reviewer 3 Report

Hello,

Thank you for addressing my comments. I feel it is much improved and should be accepted.